# The Diverse Calpain Family in Trypanosomatidae: Functional Proteins Devoid of Proteolytic Activity?

**DOI:** 10.3390/cells10020299

**Published:** 2021-02-01

**Authors:** Vítor Ennes-Vidal, Marta Helena Branquinha, André Luis Souza dos Santos, Claudia Masini d’Avila-Levy

**Affiliations:** 1Laboratório de Estudos Integrados em Protozoologia, Instituto Oswaldo Cruz, Fundação Oswaldo Cruz (FIOCRUZ), 21040-360 Rio de Janeiro, Brazil; davila.levy@ioc.fiocruz.br; 2Laboratório de Estudos Avançados de Microrganismos Emergentes e Resistentes, Instituto de Microbiologia Paulo de Góes, Universidade Federal do Rio de Janeiro (UFRJ), 21941-901 Rio de Janeiro, Brazil; mbranquinha@micro.ufrj.br (M.H.B.); andre@micro.ufrj.br (A.L.S.d.S.); 3Programa de Pós-Graduação em Bioquímica, Instituto de Química, Universidade Federal do Rio de Janeiro (UFRJ), 21941-909 Rio de Janeiro, Brazil

**Keywords:** cysteine peptidase, *Trypanosoma*, *Leishmania*, chemotherapy

## Abstract

Calpains are calcium-dependent cysteine peptidases that were originally described in mammals and, thereafter, their homologues were identified in almost all known living organisms. The deregulated activity of these peptidases is associated with several pathologies and, consequently, huge efforts have been made to identify selective inhibitors. Trypanosomatids, responsible for life-threatening human diseases, possess a large and diverse family of calpain sequences in their genomes. Considering that the current therapy to treat trypanosomatid diseases is limited to a handful of drugs that suffer from unacceptable toxicity, tough administration routes, like parenteral, and increasing treatment failures, a repurposed approach with calpain inhibitors could be a shortcut to successful chemotherapy. However, there is a general lack of knowledge about calpain functions in these parasites and, currently, the proteolytic activity of these proteins is still an open question. Here, we highlight the current research and perspectives on trypanosomatid calpains, overview calpain description in these organisms, and explore the potential of targeting the calpain system as a therapeutic strategy. This review gathers the current knowledge about this fascinating family of peptidases as well as insights into the puzzle: are we unable to measure calpain activity in trypanosomatids, or are the functions of these proteins devoid of proteolytic activity in these parasites?

## 1. Introduction

The Trypanosomatidae family, class Kinetoplastea, encompasses exclusively parasitic protozoa, some of which cause important human diseases [1]. About 37 million people worldwide are infected either with *Trypanosoma brucei,* the etiological agent of African sleeping sickness; *Trypanosoma cruzi,* the causative agent of Chagas disease; or with different species of the genus *Leishmania,* responsible for different clinical manifestations known as leishmaniasis [2]. The current therapy to treat these diseases is unsatisfying due to their low efficacy, high cost, toxicity, and tough administration routes, like parenteral [3,4]. Therefore, the search for more effective drugs is still an urgent need, which can lead to alternative strategies, such as a repurposed approach with compounds already approved for human usage [5].

In view of this scenario, calpains are an interesting target due to the intense efforts to develop means of identifying selective inhibitors in this group of calcium-dependent cysteine peptidases [6]. Since these enzymes are involved in crucial physiological roles in mammals, their deregulated activity is implicated in several pathophysiological processes, especially in fibrotic diseases and neurological disorders [7]. Moreover, trypanosomatids harbor a large and diverse family of calpain sequences in their genomes, comprising a wide range of associated domains, differential gene expression among life-cycle forms, and ubiquitous distribution in the parasite cell body [8,9,10,11]. However, due to the difficulties in assaying calpain activity in these microorganisms, there is still an open question: are trypanosomatid calpains proteolytically active?

## 2. Effects of Calpain Inhibitors against Trypanosomatid Parasites

The development of a new drug is a time-consuming, laborious, and expensive process. Considering chemotherapy to treat neglected tropical diseases, the reality is even harder, since the link with poverty results in low investment in rational drug development. Therefore, the main challenges for the introduction of new compounds to treat neglected diseases, such as the ones caused by trypanosomatids, are more economical than biological [3]. In this sense, a repurposed approach with approved drugs has potential benefits such as reducing the costs during discovery and development, preclinical laboratory tests, and clinical phases for drug safety [12,13]. For instance, the selective calpain inhibitor BLD-2660 of Blade Therapeutics Co. was approved for a phase I clinical trial to treat idiopathic pulmonary fibrosis but this compound was repositioned to treat hospitalized patients of coronavirus disease-19 (COVID-19) in a phase II trial (ClinicalTrials.gov Identifier: NCT04334460) [14].

In view of this scenario, over the past years, our research group has been advocating the usage of calpain inhibitors as an alternative chemotherapy against Chagas disease and leishmaniasis [5,15]. We started to study the effects of the calpain inhibitor MDL28170 (inhibitor III, Z-Val-Phe-CHO), which is a potent and cell-permeable inhibitor of mammalian calpains [16], against *Leishmania amazonensis*. MDL28170 decreased the in vitro proliferation of *L. amazonensis* promastigotes in a dose-dependent manner, with an IC_50_ of 23.3 µM [17]. The ultrastructural alterations observed in treated promastigotes suggested an apoptotic-like process induced by MDL28170 [18]. In addition, our research group reported that the calpain inhibitor was capable of affecting the interaction process of *L. amazonensis* promastigotes with peritoneal mouse macrophages in a time- and dose-dependent manner, as well as notably decreasing the number of amastigotes per macrophage [19]. The same effects of MDL28170 against the growth of promastigotes and on amastigote viability were also observed in *Leishmania braziliensis*, *Leishmania major*, *Leishmania infantum,* and *Leishmania donovani* [20]. In addition, ultrastructural alterations were reported in *L. braziliensis* promastigotes treated with MDL28170 that were conceivable with autophagy, and an enhanced expression of the virulence factor GP63 was observed [10]. However, although MDL28170 is a relatively specific calpain inhibitor, it cannot be ruled out that it may act on other *Leishmania* cysteine peptidases to a lesser degree [5,6,7]. Further studies on trypanosomatid calpain orthologues employing other existing drugs developed for the inhibition of human calpains should be carried out, as well as three-dimension structure elucidation and drug docking simulations.

As occurs in *Leishmania* parasites, *T. cruzi* is strongly affected by the calpain inhibitor MDL28170 in a time- and dose-dependent manner. Our research group observed that the inhibitor was capable of arresting the growth of three strains of *T. cruzi* epimastigotes, as well as increasing the expression of the virulence factor cruzipain [21]. Considering the clinically relevant forms (amastigotes and trypomastigotes), MDL28170 was also capable of significantly reducing the viability of bloodstream trypomastigotes, presenting an IC_50_ of 20.4 µM, and presented a significant reduction in the percentage of intracellular amastigotes, resulting in a diminished in vitro infection of mouse macrophages without displaying any relevant cytotoxic effect on mammalian host cells [22]. In addition, we reported the effects of MDL28170 impairing the metacyclogenesis process (the differentiation of epimastigotes to trypomastigotes in the insect vector) in a time- and dose-dependent manner, also reducing the interaction with the insect vector *Rhodnius prolixus*, and promoting several ultrastructural alterations in epimastigotes, such as the disorganization of the reservosomes, Golgi, and plasma membrane [23].

Monoxenous trypanosomatids, which by definition infect invertebrate hosts only (mainly insects), in addition to phytomonads, responsible for plant infections, have been considered as an important tool for biochemical and molecular comparative studies in trypanosomatids as well as an interesting missing piece of the puzzle to unveil trypanosomatid evolution. These organisms are used routinely because they are easily cultured under axenic conditions [24], and they contain homologues of virulence factors from the classical human trypanosomatid pathogens, such as peptidases [25]. Considering these advantages to work with monoxenous trypanosomatids, our research group studied the effects of calpain inhibitors against the endosymbiont-harboring monoxenic trypanosomatid *Angomonas deanei*, and against the tomato pathogen *Phytomonas serpens*. For *A. deanei*, three calpain inhibitors were tested and, as a result, the reversible calpain inhibitor MDL28170 displayed a higher efficacy in decreasing the in vitro parasite growth compared to the non-competitive calpain inhibitor PD150606, while the irreversible calpain inhibitor V only marginally affected the parasite growth [26]. In *P. serpens*, we observed that MDL28170 treatment of promastigotes impacted on parasite biology, such as ultrastructure, the differential expression of gp63-like, cruzipain-like and calpains, the peptidase activity, and the interaction of the parasite with the invertebrate host [27]. To further explore the possible roles of *P. serpens* calpains, an MDL-resistant strain was generated and compared to the wild type cells. The major difference observed between both strains was the presence of microvesicles within the flagellar pocket of the resistant parasites, as revealed by ultrastructural analysis [28]. Table 1 summarizes the main effects of calpain inhibitors against trypanosomatid parasites reported by our research group.

## 3. Calpain Superfamily Expansion in Trypanosomatid Genomes and Their Gene Expression

Calpains were characterized primarily in humans and, therefore, calpain homologues have been identified based on the primary sequence characteristics of the proteolytic core domain (CysPc) in almost all eukaryotes and some bacteria [7]. Consequently, the mammalian conventional calpain domain organization characterizes the reference for a calpain structure, representing the “classical” calpains, in contrast to “non-classical” ones that differ in their domain architecture. Moreover, the classification of calpains in families is constantly being updated. In general, non-mammalian organisms have few copies of these non-classical calpains in their genomes, but in trypanosomatids, a massive expansion of calpain genes took place, sometimes harboring additional domains with the most varied functions and origins [8,10,11,29].

The first screening of calpain gene expansion in trypanosomatids was performed in 2005, as soon as the *T. brucei*, *T. cruzi,* and *L. major* genomes were available. Through whole-genome analysis, Ersfeld and co-workers [8] described the presence of a diverse family of 18 calpain sequences in *T. brucei*, 24 sequences in *T. cruzi,* and 27 in *L. major*. The authors divided these sequences into five groups based on their structural domain organization. Members of groups 1 and 2 present CysPc and calpain-type beta-sandwich (CBSW) domains, being distinguished by their N-terminal domains. Members of group 3 were called small kinetoplastid calpain-related proteins (SKCRPs) since the sequences consisted only of the N-terminal kinetoplastid-exclusive domain (DUF1935). Finally, groups 4 and 5 contained highly divergent calpains enriched in repeats of their CysPc domains and/or repeat-rich (RPT) domains, respectively [8]. At this time, the authors called these calpains sequences “calpain-like proteins (CALPs)”, due to the lack of some classical calpain domains but, here, we refer to these molecules as just “calpain sequences”, like any other non-classical calpain distributed in almost all living organisms [7]. In a period of a few years, the same research group extended their studies in *T. brucei* calpains by RNA interference (RNAi) silence approach and gene expression analysis. Olego-Fernandez and colleagues [30] reported that the cytoskeleton-associated protein 5.5 (CAP5.5), the first *T. brucei* calpain-related protein discovered using biochemical techniques [31], had a paralogous copy with an analogous function called CAP5.*5V*. Both proteins have essential roles needed for correct morphogenetic patterning during the cell division cycle and for the organization of the subpellicular microtubule corset, and they also have differential gene expression between each life-cycle form. Moreover, by other comprehensive analyses of the expression patterns and subcellular localization of selected members of *T. brucei* calpains, it was shown that two transcripts were differentially expressed in bloodstream trypomastigotes, while three others were differentially expressed in the procyclic forms [9].

In *Leishmania* spp., the first reports on the presence of calpain homologues were disclosed using unbiased assays, such as transcriptomics and proteomics approaches. Before the sequencing and assembly of *L. major* genome, a microarray analysis of *L. major* promastigotes reported an upregulation of one calpain transcript in procyclic promastigotes and two in metacyclic forms during metacyclogenesis [32]. A few years later, Salotra and co-workers [33] identified genes that are differentially expressed in *L. donovani* isolated from post-kala-azar dermal leishmaniasis (PKDL) patients in comparison with those from visceral leishmaniasis by microarray technology. The authors found a 2-fold higher expression of five proteins in PKDL parasites, including an SKCRP [33]. Recently, motivated by genome sequencing and assemblage improvements, including the increased availability of trypanosomatid genomes, our research group screened the *L. braziliensis* whole genome to identify and classify calpain genes, their domain arrangements, and the gene expression patterns between procyclic and metacyclic promastigotes [10]. As a result, we identified 34 predicted calpain sequences distributed in 13 different chromosomes, with a wide range of domain architectures. The gene expression analysis by qPCR revealed five upregulated calpains in procyclic promastigotes, one upregulated sequence in the metacyclic form, and one procyclic-exclusive transcript.

In *T. cruzi*, the detection of calpain sequences was initially associated with stress conditions. Giese and co-workers [34] identified a calpain from *T. cruzi* strain Dm28c by microarray analysis. This transcript was 2.5 times more abundant in epimastigotes (insect form) under nutritional stress, a requirement for differentiation into the infective metacyclic trypomastigotes than in axenic epimastigotes growing in a rich culture medium. Some years later, in a molecular study aimed to better characterize the flagellar classical virulence factor H49 and its repeats, Galetovic and colleagues [29] revealed that H49 proteins are members of the calpain gene family of *T. cruzi*. The authors found 53 *T. cruzi* calpain sequences, eight of them carrying the 204 bp repeats of H49 proteins associated with the calpain-related sequences, which were referred to as H49/calpains. Then, our research group screened the genome of the *T. cruzi* CL Brener strain to identify and classify the calpain sequences, domain arrangements, and gene expression patterns among epimastigotes, amastigotes, and trypomastigotes [11]. Our analysis disclosed 63 calpain sequences with 14 different domain arrangements. The comparison of calpain mRNA abundance by qPCR revealed: (i) one highly expressed only in amastigotes (ii) three significantly upmodulated in amastigotes but downregulated in trypomastigotes; (iii) one upmodulated only in trypomastigotes; (iv) five significantly upmodulated in the insect form in comparison to the clinically relevant forms; (v) two without modulation between epimastigotes and amastigotes, but downregulated in trypomastigotes; and (vi) two constitutive expression sequences. Probably, this surprising expansion of calpain sequences and their differential expression between the life-cycle forms of the parasite may be correlated to distinct functions, such as the need for survival or growth within several distinctive environments, like the mammalian host and the insect vector [5,8,11]. An overview of calpain gene expression is summarized in Table 2.

## 4. Are the Trypanososmatids´ Calpains Proteolytically Active or Not?

Non-classical calpains exist not only in trypanosomatids but in almost all eukaryotes and bacteria studied so far, presenting few copies in fungi and, currently, no calpain gene has been found in the genome of any Archaea [6,34]. Most of these homologues have amino acid identities with classical calpains each ranging from <25% to >75%, and alterations of the three key catalytic amino acid residues (Cys, His, and Asn) in the CysPc domain could be found in many sequences. However, the substitutions in the catalytic triad do not always result in activity loss [35,36]. Recently, in order to evaluate the proteolytic activity of a bacteria thiol protease (*Tpr*) from *Porphyromonas gingivalis*, which has the CysPc domain, Staniec and co-workers [36] reported that *Tpr* proteolytically processes itself into active forms of 48, 37, and 33 kDa when incubated with high calcium concentrations in vitro. These proteolytic forms were incubated with a panel of calpains and cysteine peptidases substrates, but only Suc-Leu-Leu-Val-Tyr-AMC (calpain), Z-Phe-Arg-AMC (cysteine peptidases), and DQ-gelatin were hydrolyzed efficiently. Consistently, the activity was inhibited by classical cysteine peptidases and calpain inhibitors E-64, leupeptine, idioacetic acid, and the enzyme presented slower hydrolysis in low calcium concentrations [36]. The proteolytic activity of a calpain from an ancient organism suggests that non-classical calpains from unicellular eukaryotes, such as trypanosomatids, could also be active. However, calpains are tricky to detect biochemically, since they may be readily hydrolyzed by other abundant peptidases, and also due to the lack of a cleavage sequence specificity [6], which can make the measurement of the proteolytic activity not an easy task. Moreover, trypanosomatids show abundant activity of metallo- and cysteine-peptidases that could degrade minor peptidases or cover-up their detection [37].

Considering the sequence diversity of calpains in trypanosomatids, it is difficult to completely characterize these sequences and to assess their functions. However, the availability of RNA*i* in *T. brucei* made the characterization of these proteins more tangible [9,30,31,38]. Up to date, none of these studies reported any proteolytic activity by calpains in *T. brucei*, and the authors suggested that the loss of some amino acids from the catalytic triad made the calpains devoid of activity. All the findings support the idea that *T. brucei* calpains are strictly associated with structural functions, mainly acting as microtubule-stabilizing proteins with crucial roles in trypanosome morphology transitions.

Nevertheless, there are few reports disclosing an enzymatic activity in trypanosomatids peptidases that are likely to be calpains. Since the genomes of the studied organisms were not available and no attempt was made to sequence the protein or its gene, only biochemical data provided insight into the calpain characteristic of the enzyme. In 1993, Bhattacharya and colleagues [39] revealed the presence of a neutral calcium-dependent proteolytic activity in *L. donovani* promastigote extract, which was called caldonopain. More intense activity was reported at pH 7.4 before a 12 h incubation required for the caldonopain activation. The authors assumed that the delay in the activation of proteolysis in the presence of calcium ensured that this particular peptidase was a calpain. Moreover, the enzyme was inhibited by sulfhydryl reagents, such as N-ethylmaleimide and iodoacetamide, and it was found to be localized in the cytosol along with a possible specific inhibitor named caldonostatin [39]. Some years before, the same research group showed a 95 kDa protein in *L. donovani* cytosolic fraction detected in sodium dodecyl sulfate polyacrylamide gel electrophoresis (SDS-PAGE) containing gelatin, with its proteolytic activity enhanced in the presence of calcium ions [40]. A similar activity was found in amastigotes, which was drastically reduced by the addition of classical cysteine peptidase inhibitors. However, *L. donovani* peptidase(s) responsible for this activity were not purified and their identities were not disclosed.

In addition, our research group characterized a peptidase activity through the purification of an actively secreted cysteine peptidase in *A. deanei* that shares some features with calpains [41]. This cysteine peptidase was an 80 kDa homotrimer protein that exhibited maximal activity at neutral pH and was completely inhibited by the cysteine peptidase inhibitor E-64 and EGTA, which strongly suggested its calpain nature. Moreover, ion depletion completely abrogated the proteolytic activity, which was fully restored by CaCl_2_ and only partially by other ions. Although the amino acid sequence of the purified protein was not demonstrated, the *A. deanei* protein cross-reacted with antibodies raised against a calpain from *Drosophila melanogaster* [41].

Fortunately, the genome of *A. deanei* was recently sequenced and assembled [42], making feasible a screening for calpain sequences in this organism, allowing the identification of genes that keep the basic requisites to retain proteolytic activity. Therefore, here, we screened the *A. deanei* genome using previously described approaches [10,11]. Our search revealed the presence of 65 calpain sequences distributed in two of the four different assemblies available for the *A. deanei* genome (GenBank ID 14191) (Appendix A). The domain database analysis identified a wide range of different domain arrangements associated with classical calpain domains (CysPc and CBSW), similar to our reports in *L. braziliensis* [10] and *T. cruzi* [11] (Figure 1). However, here, our analysis revealed the presence of the calpain catalytic triad in seven *A. deanei* sequences (Table 3; Appendix A), while only four were detected in *L. braziliensis* and *T. cruzi* analysis. Although the identity of the proteolytic core (CysPc) of these seven calpain sequences with human m-calpain is only around 25–30%, three of them have a predicted molecular mass of around 80 kDa, the same size of the purified calcium-dependent cysteine peptidase monomer detected previously by our research group [41]. Phylogenetically, several evolutive events took place some millions of years ago, and an ancestor of the *Trypanosoma* genera was separated from *Leishmania* and *Angomonas* 231 to 283 million years ago [43]. No evolutive relationship was found among the number of calpain sequences, the domain arrangement, or the presence of the catalytic triad among the trypanosomatids species described here.

In our more recent study, no calpain activity was detected in crude extracts from *T. cruzi* epimastigotes by gelatin zymography or by fluorimetry with N-Suc-Leu-Leu-Val-Tyr substrate [10], which is consistent with the report from another research group [29]. Curiously, *T. cruzi* is the trypanosomatid with the largest expansion of the metallopeptidase gp63 in the genome with 425 sequences, but the proteolytic activity detected by zymography assays is fairly low. On the other hand, *Leishmania* spp. presents abundant metallopeptidase activity and a low number of gp63 genes, such as only 2 copies in *L. donovani* genome. It seems that there is an indirect correlation between gene expansion and proteolytic activity [44], although the evolutionary advantages/consequences or the biological impacts are still unclear.

## 5. Conclusions and Perspectives

Although massive efforts have been made to identify the proteolytic activity of trypanosomatids calpains, it still remains an open question. Are they inactive or are we unable to detect calpain proteolysis in trypanosomatids? Further efforts directed to detect possible calpain activity are necessary to help us increase our knowledge about these proteins and solve this puzzle. Some hypotheses have arisen from the knowledge of other calpain systems, mainly the mammalian ones, to explain the lack of definitive proof of calpain activity in trypanosomatids: (1) considering how fine the regulation of this enzyme must be, an endogenous highly specific inhibitor might be preventing enzymatic detection in crude extracts [45]; (2) SDS can irreversibly denature calpains, which might be preventing its identification in zymography [46] (although our attempts to use non-denaturing zymography were unsuccessful to date, data not published); (3) the abundance of other trypanosomatids endopeptidases, like gp63 and cruzipain, can quickly degrade calpains and its specific endogenous inhibitor. Up to now, trypanosomatids calpains play a structural, non-proteolytic function..

## Figures and Tables

**Figure 1 cells-10-00299-f001:**
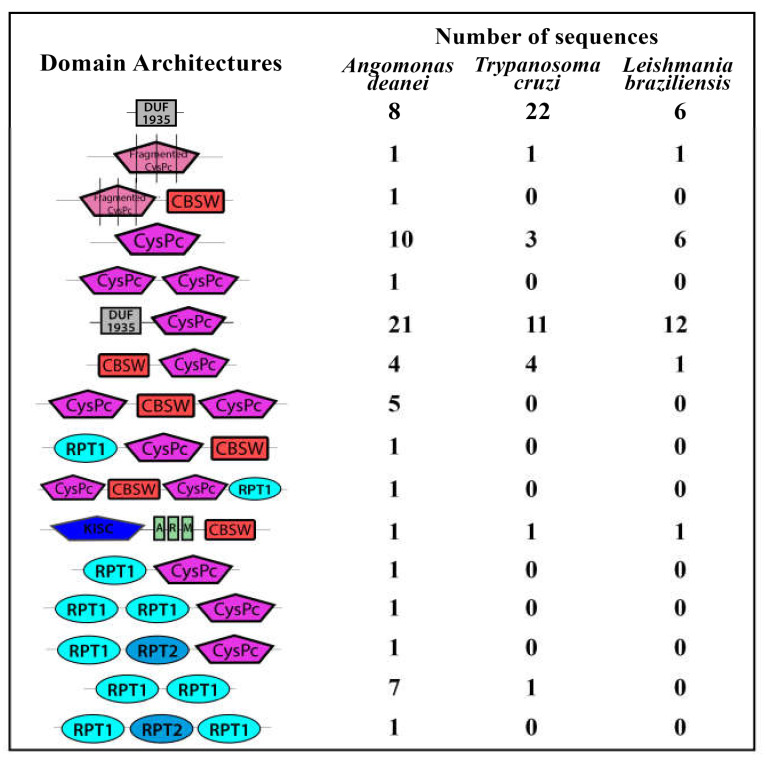
Schematic representation of the domain architectures from calpain sequences found in the *Angomonas deanei* genome. The calpain sequences were retrieved from two different assemblies from the *Angomonas deanei* genome (Genbank ID 14191) and were locally analyzed by the Simple Modular Architecture Research Tool (SMART) for the domain’s presence in InterPro and Pfam databases. The number of calpain sequences from *Trypanoma cruzi* [10] and *Leishmania braziliensis* [11] presenting the same domain architectures is illustrated for comparison. Sequences annotated as “calpain-like proteins” that have less than 100 amino acids or that do not have any calpain associated domain were not included. DUF1935—conserved N-terminal calpain-related domain from trypanosomatids, CysPc—proteolytic core domain, CBSW—calpain-type alpha-beta-sandwich domain, KISC—kinesin domain, ARM—armadillo/beta-catenin-like repeats, RPT1 and RPT2—two different repeated domains found in de-ubiquitinating proteins. Fragmented CysPc stands for short amino acid sequences from the catalytic domain.

**Table 1 cells-10-00299-t001:** In vitro effects of calpain inhibitors against trypanosomatid parasites.

Trypanosomatid	Life-Cycle Form	Compound	Observed Effect	Reference
*L. amazonensis*	promastigotes	MDL28170	Arrested irreversible growth with an IC_50_ of 23.3 μM; apoptosis-like death.	[17,18]
	amastigotes		Impaired the interaction with mouse macrophages and decreased amastigotes inside the host cell.	[19]
*L. braziliensis*	promastigotes	MDL28170	Arrested irreversible growth with an IC_50_ of 6.6 μM; increased the gp63 expression; autophagic ultrastructural alterations.	[10,20]
	amastigotes		Impaired the interaction with mouse macrophages and decreased amastigotes inside the host cell.	[20]
*L. major, L. mexicana, L. infantum, L. donovani*	promastigotes	MDL28170	Arrested irreversible growth with IC_50_ values ranging from 4.0 to 9.3 μM.	[20]
	amastigotes		Impaired the interaction with mouse macrophages and decreased amastigotes inside the host cell.	
*T. cruzi*	epimastigotes	MDL28170	Arrested growth with an IC_50_ of 34.7 μM; increased cruzipain expression; impaired metacyclogenesis; decreased interaction with the invertebrate host; ultrastructural alterations on reservosomes, Golgi, and plasmatic membrane.	[21,23]
	amastigotes		Decreased amastigotes inside peritoneal mouse macrophages.	[22]
	trypomastigotes		Reduced the viability of bloodstream trypomastigotes, presenting an IC_50_ of 20.4 μM; impaired the interaction with mouse macrophages.	[22]
*A. deanei*	choanomastigotes	MDL28170	Arrested the growth with an IC_50_ of 64.4 μM for the wild type strain and 51.3 μM for the aposymbiotic strain.	[26]
		PD150606	Arrested the growth with an IC_50_ of 231.6 and 248.3 μM for the wild type and aposymbiotic strains, respectively.	
		inhibitor V	Slightly decreased the growth of both strains.	
*P. serpens*	promastigotes	MDL28170	Arrested the growth with an IC_50_ of 30.9 µM; ultrastructural alterations of rounding of the parasite cell body, cell shrinkage, loss or shortening of the flagellum, and mitochondrial swelling; increased the expression and activity of cysteine peptidases; promoted microvesicular formation within the flagellar pocket of a resistant strain.	[27,28]

**Table 2 cells-10-00299-t002:** Gene expression findings of trypanosomatid calpains.

Trypanosomatid	Life-Cycle Form	Method	Result	Reference
*L. donovani*	post-kalazar amastigotes	microarray	Upmodulation of one SKCRP	[33]
*L. major*	meta- and procyclic promastigotes	microarray	Two upmodulated calpains in metacyclics and one in procyclic forms	[32]
*L. braziliensis*	meta- and procyclic promastigotes	qPCR	Five upmodulated calpains in procyclics, one upmodulated in metacyclic forms, and one procyclic-exclusive transcript	[10]
*T. brucei*	procyclic and bloodstream trypomastigotes	qPCR	Life-cycle-specific expression of CAP5.5 (procyclics) and its analogous CAP5.5*V* (bloodstream forms)	[30,31]
			Three upmodulated calpains in procyclics and two in bloodstream trypomastigotes	[9]
*T. cruzi*	epimastigotes, amastigotes, and trypomastigotes	qPCR	One highly upmodulated calpain in amastigotes; three upmodulated in amastigotes but downmodulated in trypomastigotes; one upmodulated in trypomastigotes; five upmodulated in epimastigotes; two downmodulated in trypomastigotes; and two constitutive calpains	[11]

**Table 3 cells-10-00299-t003:** Calpain sequences with the conserved catalytic triad (C,H,N) from *Angomonas deanei*.

Sequence ID	Domain Architectures	Predicted Molecular Masses (kDa)	Identity with Human m-Calpain CysPc
EPY31064.1	CysPc	61.90	26.71
EPY40095.1	CysPc	66.74	28.66
EPY35473.1	DUF1935, CysPc	81.01	26.09
EPY27608.1	DUF1935, CysPc	94.24	28.80
EPY32139.1	DUF1935, CysPc	80.20	28.66
EPY35937.1	DUF1935, CysPc	80.33	28.66
EPY41780.1	CysPc, CBSW	123.76	21.90

Calpain sequences were retrieved from the *A. deanei* genome. The domain combinations were obtained from the Simple Modular Architecture Research Tool (SMART). The predicted molecular mass was calculated in Bioinformactis. org. CBSW—calpain-type alpha-beta-sandwich domain, CysPc—proteolytic core domain, DUF1935—conserved N-terminal calpain-related domain from trypanosomatids. The identity with human m-calpain CysPc (NP_001739.3) was obtained by the sequence alignment in Clustal Omega.

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
