# Peer review of "The Diverse Calpain Family in Trypanosomatidae: Functional Proteins Devoid of Proteolytic Activity?"

_cells, 2021, doi:10.3390/cells10020299_

Round 1
Reviewer 1 Report
The manuscript by Ennes-Vidal et al. nicely summarizes current information about calpain and/or calpain-related proteases found in Trypanosomatidae. The authors survey the current knowledge about calpain species found/identified/characterized in several parasites (chapter 1,2,3 and the first half of 4) and describe some interesting properties that they found by looking into genome assembly of A. deanei (the lest part in chapter 4). Considering the current world health situation, it is worthy to keep raising awareness for infectious diseases and the potential targets therein. Therefore, revising the manuscript accordingly to the following suggestion should be considered.
- Contents described in the chapter 4 (identification and structure comparison) will be better appreciated with a figure rather than a table. Plus, what is common and what is not among A. deanei, L. braziliensis, and T. cruzi, should be schematically represented.
- The last paragraph of the chapter 4 (Although huge efforts …, line 287), should be composed as an independent chapter and include perspective for the future direction of this research field.
- Substrate specificities seem to be a big issue in this protease family. What are described in lines 278-286 should be better edited since the authors refer to similar problems in lines 209-212.
- To summarize the numbers and structures of calpain-related genes and their protein products described in the manuscript, additional figures should be presented.
- It will be appreciated by researchers in other fields to schematically show evolutionary relationships among (closeness or distance) among Trypanosomatid species described in this manuscript.
Author Response
Reviewer 1:
The manuscript by Ennes-Vidal et al. nicely summarizes current information about calpain and/or calpain-related proteases found in Trypanosomatidae. The authors survey the current knowledge about calpain species found/identified/characterized in several parasites (chapter 1,2,3 and the first half of 4) and describe some interesting properties that they found by looking into genome assembly of A. deanei (the lest part in chapter 4). Considering the current world health situation, it is worthy to keep raising awareness for infectious diseases and the potential targets therein. Therefore, revising the manuscript accordingly to the following suggestion should be considered.
Authors: Dear reviewer, first of all, we would like to thank you for the careful analysis and criticism, which were used to improve the MS. A highlighted MS version containing the modified parts were upload to facilitate your analysis and a point-by-point answer to your comments/concerns can be found below.
Reviewer 1:
- Contents described in the chapter 4 (identification and structure comparison) will be better appreciated with a figure rather than a table. Plus, what is common and what is not among A. deanei, L. braziliensis, and T. cruzi, should be schematically represented.
Authors: We appreciate your remark and agree that a figure could be better appreciated. The difficulties are related to the number and diversity of sequences and also to the fact that Angomonas deanei genome is not curated. Consequently, we have a massive number of the most varied domain architectures to schematize, which makes the figure representation hard. Nevertheless, we believe that the contents described in chapter 4 is now better visualized by a new figure, containning the domain arrangements (Figure 1) and an alignment of CysPc domains from the sequences with the conserved catalytic triad (Figure S1. Please find the Figure 1 in page 8 from the reviewed version of the MS.
Reviewer 1:
2. The last paragraph of the chapter 4 (Although huge efforts …, line 287), should be composed as an independent chapter and include perspective for the future direction of this research field.
Authors: We appreciate your suggestion and included a new chapter “Conclusions and perspectives” in the last paragraph. Thanks for pointing it out.
Reviewer 1:
3. Substrate specificities seem to be a big issue in this protease family. What are described in lines 278-286 should be better edited since the authors refer to similar problems in lines 209-212.
Authors: We appreciate your criticism and agree that the substrates used should be described. Please, find it in lines 303-304 and 306-307.
Reviewer 1:
4. To summarize the numbers and structures of calpain-related genes and their protein products described in the manuscript, additional figures should be presented.
Authors: In fact, figures could summarize and illustrate better the trypanosomatids calpain sequences. We hope that the new “Figure 1” improved the manuscript interpretation.
Reviewer 1:
5. It will be appreciated by researchers in other fields to schematically show evolutionary relationships among (closeness or distance) among Trypanosomatid species described in this manuscript.
Authors: We appreciate your comments and thanks one more time for the careful analysis of our MS. The trypanosomatids’ family, as well as many others protozoa families, are very diverse and several evolutive events took place some millions of years ago (reviewed by Lukes et al., 2014). The ancestral of the Trypanosoma genera was separated from the Leishmania and Crithidia 231 to 283 million of years ago. However, up to date, we didn’t find any relationship between the number of calpain sequences, the presence of the catalytic triad or the different domain arrangements between the trypanosomatids species. In a curated databank TriTryDB, for example, there are 26 to 145 sequences annotated as calpain in the genome of the different species, without any association of sub-family or genera. In the included Figure 1 of our reviewed version, it is possible to observe that one domain organization found in A. deanei could be found in T. cruzi but not in L. braziliensis, which is more closely related. Therefore, although tempting, a phylogenetic relationship between the calpain sequences is not an easy task to represent. We added some considerations about the evolutive relationship about the three trypanosomatids described in lines 276-279 and a new reference (Lukes et al., 2014) to improve the MS.
Reviewer 2 Report
Summary
Proteins with the signature calpain catalytic domain have greatly expanded in trypanosomes suggesting a possible target for specific pharmaceuticals to attack this clinically important parasite. There is a review of the literature on effectiveness and actions of distinct Calpain inhibitors on a number of trypanosome species. The review then discusses bioinformatic studies on calpains on trypanosomes and several molecular studies where calpain or calpain-like molecules were inactivated by RNA interference. The expression of calpains in distinct stages of trypanosomes is also discussed. This is followed by a discussion of the difficulties in ascertaining whether the trypanosome calpains are catalytically active or not. Overall, the review comprehensively covers what is known about trypanosome calpains.
Critique
While the review is in general accurate, there are a few areas that need improvement.
Calpain family. The review talks only about classical and non-classical calpains and uses the classical calpain as the “reference” for calpain structure (Line 122). However, from an evolutionary point of view this does not make much sense. A number of bioinformatic papers on calpain evolution have defined five main families of animal calpains, and the non-classical PALB family is the most ancient as it is present in Fungi and animals (Jekely and Friedrich, 1999; Hastings et al, 2017). It is also stated that there are “few copies of these non-classical calpains in non-mammalian organisms (Line 125). This is not true as the non-classical calpain families are conserved throughout the animal kingdom with numbers being equal to or higher than in mammals (Rawlings, 2015). Several atypical and classical calpain expansions in non-mammalian organisms have been characterized including in nematodes (Joyce et al, 2012) and molluscs (Hastings et al, 2017). The section on the calpain family needs to be updated.
Jékely G, Friedrich P. The evolution of the calpain family as reflected in paralogous chromosome regions. J Mol Evol. 1999;49:272–81.
Rawlings ND. Bacterial calpains and the evolution of the calpain (C2) family of peptidases. Biol Direct. 2015; 10:66. https://doi.org/10.1186/s13062-015-0095-0 PMID: 26527411
Hastings MH, Gong K, Freibauer A, Courchesne C, Fan X, Sossin WS (2017) Novel calpain families and novel mechanisms for calpain regulation in Aplysia. PLoS ONE 12(10): e0186646. https://doi.org/10.1371/journal.pone.0186646
Joyce PI, Satija R, Chen M, Kuwabara PE. The atypical calpains: evolutionary analyses and roles in Caenorhabditis elegans cellular degeneration. PLoS Genet. 2012;8(3):e1002602. doi:10.1371/journal.pgen.1002602
The review does not seem to consider that whether trypanosome calpains are active is an important consideration for whether calpain inhibitors that have effects on trypanosomes are actually having their impact through calpains. This needs a more thorough discussion to better unify the review; clearly if the calpains are inactive, then the target of the inhibitors is a distinct protease. This is also relevant for the justification for the main calpain inhibitor used (MDL28170). How was this inhibitor designed? what is its specificity within and outside the calpain family? What is the likelihood that it is inhibiting a target that is not a calpain?
Grammatical issues:
Line 287 “huge efforts”
“huge” is not an appropriate scientific term.
LINE 42: calpains call attention
LINE 54: Considering the neglected diseases research
Line 101: In this sense, our research group 101 studied (the antecedent is not clear)
Line 298-299. Yet to be proven, for the moment, trypanosomatids calpains play a structural non-proteolytic role.
Author Response
Reviewer 2:
While the review is in general accurate, there are a few areas that need improvement.
Calpain family. The review talks only about classical and non-classical calpains and uses the classical calpain as the “reference” for calpain structure (Line 122). However, from an evolutionary point of view this does not make much sense. A number of bioinformatic papers on calpain evolution have defined five main families of animal calpains, and the non-classical PALB family is the most ancient as it is present in Fungi and animals (Jekely and Friedrich, 1999; Hastings et al, 2017). It is also stated that there are “few copies of these non-classical calpains in non-mammalian organisms (Line 125). This is not true as the non-classical calpain families are conserved throughout the animal kingdom with numbers being equal to or higher than in mammals (Rawlings, 2015). Several atypical and classical calpain expansions in non-mammalian organisms have been characterized including in nematodes (Joyce et al, 2012) and molluscs (Hastings et al, 2017). The section on the calpain family needs to be updated.
Jékely G, Friedrich P. The evolution of the calpain family as reflected in paralogous chromosome regions. J Mol Evol. 1999;49:272–81.
Rawlings ND. Bacterial calpains and the evolution of the calpain (C2) family of peptidases. Biol Direct. 2015; 10:66. https://doi.org/10.1186/s13062-015-0095-0 PMID: 26527411
Hastings MH, Gong K, Freibauer A, Courchesne C, Fan X, Sossin WS (2017) Novel calpain families and novel mechanisms for calpain regulation in Aplysia. PLoS ONE 12(10): e0186646. https://doi.org/10.1371/journal.pone.0186646
Joyce PI, Satija R, Chen M, Kuwabara PE. The atypical calpains: evolutionary analyses and roles in Caenorhabditis elegans cellular degeneration. PLoS Genet. 2012;8(3):e1002602. doi:10.1371/journal.pgen.1002602
Authors: We would like to thanks the reviewer for your careful analysis and your concerns. In fact, the calpain classification is very extensive and there are many manuscripts (MS) dedicated to better elucidate such classification. We are very grateful for the listed reviews suggested and apologized by our omission about the, at least, five calpain families. In this MS, we summarized few introductory concerns about the calpain classification and structures based on the review from Ono and co-workers (2016), where the suggested references were summarized. Our group was invited to write a MS about calpains from trypanosomatids in this special issue and decided to make a mini-review addressing the puzzle about calpain activity in these microorganisms. Therefore, we believe that a more extensive revision about calpain structures and classification from other organism and its comparison with trypanosomatids’ calpains could be better described by more specialized researchers, probably in this same special issue. Nevertheless, it does not mean we will keep wrong statements. In order to improve the final MS, we make some improvements in our short introduction about calpains in lines 130-132 of the reviewed MS.
There is also a misunderstanding in the mentioned line 125 of the submitted MS. We apologized for the statement, but our intention was to state that there are few copies of calpains in the genome of each non-mammalian organism in comparison of the great expansion of calpain sequences in trypanosomatids. Please, find a modified sentence to improve this statement highlighted in lines 131-132 of the new MS version.
Reviewer 2:
The review does not seem to consider that whether trypanosome calpains are active is an important consideration for whether calpain inhibitors that have effects on trypanosomes are actually having their impact through calpains. This needs a more thorough discussion to better unify the review; clearly if the calpains are inactive, then the target of the inhibitors is a distinct protease. This is also relevant for the justification for the main calpain inhibitor used (MDL28170). How was this inhibitor designed? what is its specificity within and outside the calpain family? What is the likelihood that it is inhibiting a target that is not a calpain?
Authors: It´s a very interesting and intriguing question extensively debated in our previous reports. When our studies about the effects of calpain inhibitors started there were few reports about the unspecific effects of the calpain inhibitor III MDL-28170, and it was the compound which presented the most relevant effects against trypanosomatids. Unfortunately, until nowadays we don´t have a tridimensional structure of trypanosomatids’ calpains, as well as many others proteases, to analyze the molecular docking by bioinformatic approaches to elucidate such specificities and interactions. Also, in spite of all the efforts we have made, the measurement of the activity of trypanosomatids’ calpains is not an easy task, therefore, we are not able to determine to what extent MDL28170 is affecting calpains directly or other related cysteine peptidases. However, ultrastructural analysis of MDL-28170 effects against T. cruzi parasites reported damages in cellular organelles where calpains were abundantly detected by proteome approaches, such as revervosomes (Santana et al., 2008) and membrane (Cordeiro et al., 2009).
Nevertheless, these data don´t rule out the possibility of an unspecific effect of MDL-28170 and we agree that this compound could be also acting on other cysteine proteases. Therefore, we added some considerations about MDL28170 in lines 81-85 of the reviewed version, which were not fully explored here due to the distinct scope of this review and because it has been already discussed in our published papers and reviews (d’Avila-Levy et al., 2006; Sangenito et al., 2009; Branquinha et al., 2013; Marinho et al., 2017; Ennes-Vidal et al., 2011, 2017 and 2019).
Reviewer 2:
Grammatical issues:
Line 287 “huge efforts”
“huge” is not an appropriate scientific term.
Authors: We apologized for the mistake and changed the word, as suggested. Thanks.
Reviewer 2:
LINE 42: calpains call attention
Authors: Thanks, changed to “are an interesting target”.
Reviewer 2:
LINE 54: Considering the neglected diseases research
Authors: We modified the sentence to better clarify this issue.
Reviewer 2:
Line 101: In this sense, our research group 101 studied (the antecedent is not clear)
Authors: The antecedent was modified, as requested.
Reviewer 2:
Line 298-299. Yet to be proven, for the moment, trypanosomatids calpains play a structural non-proteolytic role.
Authors: The sentence was modified according with the requestion.
Finally, we would like to thanks one more time for the careful analysis and for all the reviewers’ concerns. We hope our responses and improvements in the reviewed MS meet with your approval.